# Insights from COVID-19: Reflecting on the Promotion of Long-Term Health Policies in China

**DOI:** 10.3390/ijerph20042889

**Published:** 2023-02-07

**Authors:** Qi Wu, Beian Chen, Jianping Zhu

**Affiliations:** 1Data Mining Research Center, Xiamen University, Xiamen 361005, China; 2School of Management, Xiamen University, Xiamen 361005, China; 3School of Management, Zhejiang University, Hangzhou 310058, China

**Keywords:** Healthy China 2030, China, smart healthcare, COVID-19, understanding

## Abstract

China announced the Healthy China Initiative (2019–2030) in 2019, an action program aimed to support the country’s current long-term health policy, Healthy China 2030, which focuses on public health promotion and health awareness. Following the implementation of the policy, China had the COVID-19 pandemic, which had an influence on both the public’s degree of health awareness and the adoption of the HCI. This research examines whether the COVID-19 epidemic has increased public understanding and acceptance of China’s long-term health policy. In addition, it analyzes whether the Chinese public’s awareness of health policy has been impacted by China’s usage of smart healthcare in its response to the pandemic. To correspond to these study aims, we used a questionnaire based on the research questions and recent relevant research. The results of the study, based on an examination of 2488 data, demonstrate that the Healthy China Initiative is still poorly understood. More than 70% of respondents were unfamiliar with it. However, the results imply that respondents are becoming more aware of smart healthcare and that public acceptance of official health policies can be aided by the sharing of knowledge about this. As a result, we examine the situation and draw the conclusion that the spread of cutting-edge health-related technology can enhance the communication of health policy and provide participants and policymakers with fresh insights. Finally, this study also can provide lessons for other countries in the early stages of policy dissemination, particularly health policy advocacy and promotion during epidemics.

## 1. Introduction

On 15 July 2019, the Chinese government launched the latest implementation plan for long-term health policy, the Healthy China Initiative (HCI). This policy statement clarifies the goals and action plan of HCI, which aims to shift the core direction of China’s long-term health policy from disease control to disease prevention over the next decade [1]. In the literature review, a more comprehensive introduction to HCI will be provided. It is significant to notice that this strategy places a high priority on policy promotion and health literacy, both of which are included in its action plans. Additionally, policy promotion and communication are regularly underlined in public government materials. The evidence suggests that the long-term health strategy of China has made policy advocacy and the transmission of health information major components [2]. Therefore, this study took the opportunity to assess further the effects of promoting long-term health policy and look into the short-term impact and acceptability of HCI a year after it was promoted.

Early in 2020, the COVID-19 epidemic started, and it now presents a serious threat to worldwide public safety [3]. Since the beginning of the pandemic, many countries have adopted restrictive measures and imposed embargoes in an effort to contain the spread of the virus. The pandemic and these restrictive measures have changed living conditions across countries and age groups, affecting social life and increasing mental health problems on the one hand [4], which may even cause people to grieve the loss of a normal life [5], and exacerbating public health inequalities on the other hand [6]. COVID-19 also caused a significant threat to public health in China; however, China quickly and effectively controlled the rise in cases after facing a high number of infections during an initial 2-month period [7].

From mid-April 2020 onward, the number of cases in China stopped increasing, and new cases were mainly imported from outside the country [8]. The incorporation of smart medicine in China’s response to COVID-19 appears to have directly increased public acceptance and familiarity with the technology [9]. Due to the intimate relationship between smart medical technology and HCI, we believe that the Chinese public will have a better understanding of health policy as a result. So, we conducted this subsequent study to test our assumption. Our research, conducted in May 2020, intended to understand how policy promotion was functioning almost a year after the introduction of the HCI. We also investigated whether or not China leveraged its ability to control the pandemic while promoting health policy.

Therefore, our research objectives explored the following four research questions. First, what is the level of public awareness of HCI in China following the COVID-19 pandemic. Second, how familiar is the Chinese public with smart healthcare in the wake of the COVID-19 pandemic. Third, whether the level of understanding of smart healthcare can help promote understanding of long-term healthcare policy in China. Finally, how other channels can contribute to the diffusion of long-term health policies in the wake of the COVID-19 pandemic.

## 2. Literature Review

### 2.1. Long-Term Health Policy Promotion

Most nations have their own long-term health policies and programs, which are created and promoted differently based on the unique circumstances of each nation. The UK does not have a nationally consistent long-term health policy or a mandatory uniform promotion strategy; instead, individual long-term health policies are determined by each region. For example, the National Health Service (NHS) has developed the Long-Term Plan (LTP) in England. This policy has long-term objectives and is committed to enhancing policy promotion [10]; however, Miles and Asbriges [11] recognize that the policy needs to be further promoted and explained to the public. Adequate interpretation and dissemination of the policy, notably by medical practitioners, can increase professional regulation and treatment alternatives.

In contrast to the UK, which designed a relatively independent and flexible long-term policy, the US has implemented state-level health policies at 10-year intervals since 1980 through its Healthy People program. The latest policy contained 355 core or measurable objectives [12]. Moreover, long-term health policies in the United States are promoted through a variety of means, such as direct intervention in public behavior through legal and regulatory requirements [13] and small grants that incentivize local social organizations and agencies to promote and disseminate policies [14].

Similar to the United States, Japan has a long-term health policy called ‘Health Japan’ that is created every ten years. The plan clearly defines the obligations that different levels of government have to promote the policy [15]; however, as Tsuji [16] has emphasized, there are limitations with the policy’s promotion strategy, and the Japanese community still needs to be made more aware of its main objectives.

As this discussion showcases, many countries have their own long-term health policies, but they utilize varying types of promotion strategies. Our study will explore the strategies that have been used to promote long-term health policy in China, in particular, and in doing so, we will provide insights for future research.

### 2.2. Development and Promotion of the Healthy China Policy

The development of China’s current long-term health policy started in 2009 with the Healthy China 2020 program, which aimed to construct an accessible, affordable, and efficient health system in the country by 2020 [17]. The program established a large number of objectives in disease prevention and the promotion of a healthy lifestyle. However, as evidenced by Table 1, the program’s objectives lacked quantitative indicators and practical details and therefore seemed similar to slogans than meaningful policy goals [18]. Moreover, there were problems with the campaign used to promote it among the public. As Zhu Chen, the former acting Minister of Health observed, a lot of important information was not communicated to the target population of the program. One reason for this is that there was a lack of collaboration between the different departments responsible for it. The Healthy China 2020 programme thus had several limitations. However, it provided useful lessons and insights for the next phase of China’s long-term health policy.

In 2016, the Chinese government released the Healthy China 2030 Planning Outline, which was the first instance where China prioritized public health as a national strategy [2]. Public health and well-being have become an integral aspect of China’s legislative framework, and China’s health policy will be based on these factors [19]. The core aims of the blueprint have shifted from the Healthy China 2020 program’s emphasis on disease control to disease prevention and a desire to ultimately improve the health and well-being of the Chinese population through this 10-year initiative [20]. The development of the policy draws on lessons learned from previous policies and includes five specific objectives to assist in achieving the core aim (see Table 1). Moreover, on 15 July 2019, three years after the outline was announced, the Chinese State Council issued a corresponding action plan called the Healthy China Initiative. This action plan converts the five primary objectives of the outline into three tasks and 15 quantifiable, specific targets.

The first target of the HCI focuses on promoting health policy and disseminating health information among the Chinese public. This target underpins the entire initiative by encouraging adherence to its objectives [1]. To ensure the target is achieved, HCI established an independent committee with responsibility for the planning and development of a promotional strategy. This committee collaborates with Health News, the official media of the National Health Commission (NHC, formerly the Ministry of Health), to publicize health information. It is also responsible for the local government’s promotion of the HCI, encouraging widespread participation in the initiative across the country. In addition, our research reveals that, contrary to previous strategies relying on government announcements and traditional media, the NHC has registered official accounts on leading Chinese social media platforms, such as ShakeYin, Weibo, and WeChat, in order to popularize the HCI with an average of over a million users following each individual account.

The Chinese government has placed great emphasis on ensuring public understanding of the HCI and enhancing people’s knowledge and awareness of health matters. Our research will assess if the project has yielded sufficient results one year after its inception.

### 2.3. Impact of Health Policy on Public Mental Health during the Pandemic

Pandemics have become a source of stress, including threats to human life and social disruption as people are forced to change their habits and patterns of behavior [21]. Although this event is unprecedented in terms of scale, we know that these types of catastrophic events are known to affect mental health and lead to higher levels of depression [22]. A recent research analysis by Sebri et al. [23] estimates that the emotional responses of Italians during the COVID-19 outbreak highlighted moderate levels of worry, anxiety, and distress. Ettman et al. [24] concluded similarly, reporting a threefold increase in depressive symptoms in the US adult population. Furthermore, the changes to normal life have become a source of anxiety due to uncertainty about the length of the quarantine, anger over the loss of control, fear of death, illness, loss of employment, economic instability, loss of loved ones, discontent with the Spanish government, transparency, a sense of loneliness and, ultimately, fear of the unknown [5].

In addition, several unique features of China’s COVID-19 epidemic model and its management policies have contributed to the exacerbation of the public mental health crisis. First, many Chinese residents still remember the harmful effects of the 2003 outbreak of severe acute respiratory syndrome (SARS), and COVID-19 is more transmissible than SARS [25]. The uncertain incubation period and possible asymptomatic transmission of the virus caused additional fear and anxiety. Secondly, the government initially downplayed the severity of the epidemic, eroding public trust in the transparency and competence of government decision-making. Third, the unprecedented large-scale quarantine measures in all major cities, which essentially confined residents to their homes, may have had a negative psychosocial impact on residents [26]. Finally, excessive (mis)information in social media poses a significant risk to public mental health in this health crisis [27].

In line with the strategic objectives of the HCI, and based on recent experience, the National Health Commission of China issued a circular on 26 January 2020 providing guidelines for emergency psychological crisis interventions to reduce the psychosocial impact of the COVID-19 outbreak [28]. The circular specifies that psychological crisis intervention should be part of the public health response to the COVID-19 outbreak. However, the COVID-19 pandemic led to greater mental stress among the Chinese public. China’s timely provision of health policies-maintained resilience for the majority of the population even during the pandemic [29]. Similarly, Ding et al. [30] investigated the mental health and behavioral responses of Chinese adults during the COVID-19 crisis and showed that support for prevention and control policies was negatively associated with depression among people in a public health crisis. The role of public health policies in maintaining and promoting mood is thus evident, while public health policies are part of long-term health policies, making it necessary to explore how long-term health policies can be promoted.

At the same time, on the one hand, the uncertainty of a pandemic can threaten health and safety and thus lead to fear [31], but long-term health policies can be developed to alleviate the uncertainty caused by a pandemic and thus the anxiety caused by public fear [32,33]. On the other hand, COVID-19 exposes our persistent social, economic, and political inequalities. A post-COVID-19 global recession seems likely, which could worsen health equity, so a long-term health policy response is essential so that the COVID-19 epidemic does not exacerbate health inequalities for future generations [6]. However, there have been few studies on the situation in developing countries. In this paper, we focus on the short-term impact and acceptance of long-term health policy HCI on the Chinese public in a pandemic context and further analyze how to promote the benefits of long-term health policy.

### 2.4. Relationship between Health Policy and Smart Healthcare during the Pandemic

Under six months after the launch of the HCI, COVID-19 emerged in China and caused widespread infection among the population. China successfully controlled the outbreak over a short period of time, reflecting its capabilities and experience in disease control. The effectiveness of the country’s response to COVID-19 has also been linked to its use of smart healthcare to monitor the spread of the virus. Introduced in 2009, smart healthcare uses sensors to detect information related to health and employs computers with cloud technology to process it [34]. The main functions of smart healthcare include: assisting with diagnosis and treatment, health management, disease prevention, risk monitoring, and assisting drug research [35].

In the early stages of the pandemic, China applied smart health technology to assist health authorities and members of the public in identifying people infected with COVID-19. For example, the Chinese government introduced an app called Query about the Same Itinerary as a Patient, which updates information about public transport to inform passengers if they have shared a journey with someone infected with the disease. This smart healthcare technology contributed to the reduction in viral transmission in areas with COVID-19 cases [36]. Additionally, the use of artificial intelligence along with computerized tomography (i.e., X-rays used to make 3-D images) has assisted doctors in rapidly and accurately diagnosing COVID-19 while also saving medical resources [37].

It follows that China has always had detailed policies for the public as a guide to reduce the uncertainty posed by threats through the spread and application of smart healthcare technology. The nature of uncertainty in this context is that uncertainty exists when the details of the policy are vague, complex, and unpredictable and when people feel insecure about their state of knowledge or the state of knowledge in general [38,39]. The integration of smart medicine into China’s response to COVID-19 seems to have directly increased public acceptance and familiarity with the technology [9]. In the sections below, we will consider how its use during the pandemic has affected public awareness of health policy and changed the way in which it is promoted.

## 3. Materials and Methods

### 3.1. Data and Participants

#### 3.1.1. Data Source

Our research aims to assess the Chinese public’s understanding of health policy after the first effective containment of COVID-19 in China on 1 May 2020 [40], assuming that distinct populations will respond differently to health policy and that smart healthcare will have an impact on the extent to which it is understood. In order to answer our research questions, we used a cross-sectional survey design.

Our research was specifically conducted to understand the Chinese public’s perceptions of the HCI following the initial outbreak of COVID-19. We thus decided to carry out data collection over the space of a week in May 2020, when the rate of new cases of the virus had stabilized in China. In fact, that month, there were less than 60 new infections in mainland China, and of those newly reported cases, all were associated with cases imported from abroad [41]. The specific time frame of our data collection was from 12 May to 18 May.

In order to obtain data for our survey, we decided to conduct a questionnaire on WeChat. WeChat is a popular social application in China with over one million monthly active users; about 63 million users over 55 years old actively used WeChat on a monthly basis in 2019, based on Tencent’s report [42]. Using it enabled us to carry out our questionnaire without outside interference and to obtain enough data to answer our research questions [43]. From 12–18 May 2020, self-designed questionnaires on HCI perceptions were sent from the author’s WeChat account to 3000 users and WeChat groups (approximately 20 chat groups of 10,413 people). Each recipient was asked to forward the link to the questionnaire to their own network of friends. A total of 2691 questionnaires were collected, of which 2488 were determined to be valid, based on the latest demographic information published in the China Statistical Yearbook 2020 regarding factors such as gender, age, income, and region. Table 2 shows the breakdown of the demographic information of the participants, and the specific data can be found through the webpage (https://github.com/Winckwu/Reflecting-on-HCI-in-China (accessed on 1 January 2023)).

#### 3.1.2. Questionnaire Preparation and Survey Method

Questionnaires are created, distributed, and collected using the online survey tool Sojump (http://www.sojump.com, (accessed on 1 January 2023)). Sojump is a professional online survey, evaluation, and polling platform that allows for a user-friendly service, including questionnaire design, data collection, customized reporting, and analysis of results. The questionnaire entitled “questionnaire on knowledge, attitude, and behavior of Healthy China Initiative via WeChat” was by authors based on a literature review [44,45], the needs of this study and the researchers’ experiences. As a result, six main themes were developed in the questionnaire: demographic indicators, the current state of health education, knowledge of smart healthcare, knowledge of health policies, desired health education methods, and questions related to access to health information through information technology. A series of measurement indicators were then created, including channel, frequency, overall rating, and so on. These indicators on the use of information technology to access health information were divided into three levels from large to small: type, means, and mode. Finally, we set various questions, which included short answer, multiple choice, and scoring questions, and introduced 5-Likert scales and percentages for quantification, and the specific questions can be found in Appendix A. This kind of scale has been used extensively in research to measure people’s perceptions and attitudes toward policy. We use anonymity to protect the personal privacy of our respondents, so there is no conflict of ethical risk.

In the questionnaire, our six single-choice questions were related to degree. First, respondents rated the extent to which they understood the HCI. Then, they rated their understanding of the overall concept of smart healthcare. Our research posed four specific questions related to this concept to verify the consistency of respondents’ understanding of it. At the end of the questionnaire, we included four open-ended questions to understand issues related to the promotion of health policy. We asked respondents: what channels they used to obtain advice on health matters; what illnesses or other health problems they were most concerned about; what they considered to be the most important medical issues at the time; and through what means they thought the state could develop the promotion of health policy.

Once the questionnaire had been designed, five participants, two of whom were survey experts, conducted the pilot test. The questionnaire was then reworked by the research team to address issues that arose during the pilot—these issues related to response time, question design, question order, and question content. At the same time, in order to improve the quality of the returned questionnaires and to collect as many valid questionnaires as possible, electronic bonus packs of varying value were randomly distributed as an incentive for the respondents’ initiative; the actual value was 1 RMB (0.14 euros) per person.

#### 3.1.3. Statistical Analysis

The questionnaires were analyzed to exclude invalid questionnaires. Exclusion criteria included duplicate IP addresses, logical errors, identical answers to consecutive questions, and blank questions. Finally, a total of 203 questionnaires were excluded, yielding 2488 valid questionnaires. Data analysis was conducted using Sojump (Changsha Ranxing Information Technology Co., Ltd., Changsha, Hunan, China) and Excel (Microsoft, Redmond, Washington, DC, US), SPSS 20.0.0 (IBM, Armonk, New York, NY, US). Descriptive analysis was carried out automatically through the statistical function of Sojump. Finally, a least square regression analysis was conducted through SPSS to explore the relationship between smart healthcare and awareness of HCI.

For the four open-ended questions, where the answers of these respondents were very difficult to read and evaluate, we automated the evaluation of open-ended answers by constructing word clouds, a model that uses visually appealing and appropriate semantics to partially automate free-text evaluation [46]. The specific word cloud generation steps are:

First, respondents’ responses to each question were submitted and divided into sentences with several phrases, such as “not familiar with the process”, “health insurance is not universal”, etc.

Second, a numerical weight was assigned to each phrase based on how often it appeared. The formula is simple: weight = the number of occurrences.

Third, the font size is assigned in proportion to its weight, considering the size of the various constants.

Fourth, estimate the total area enclosed by the word cloud, centered on the following factors: the arch box for each word, summarizing and adjusting the area for smaller and larger phrases.

Finally, the phrases with higher frequency/weight were placed closer to the center, all in different rectangular boxes.

### 3.2. Variables

The dependent variable chosen for our research is the extent to which the Chinese public understands China’s current long-term health policy. Our respondents rated their awareness of the policy on the Likert scale, from 1 (no hear) to 5 (very understand), resulting in an ordered but non-linear variable.

The dependent variable chosen for our research is the extent to which the Chinese public understands China’s current long-term health policy. Our respondents rated their awareness of the policy on the Likert scale, from 1 (no hear) to 5 (very understand), resulting in an ordered but non-linear variable.

In order to manage the influence of demographic factors on our findings, we included gender, education, age (people’s stated age and placed into categories), place of residence, and household income as control variables related to demographics. Additionally, we considered the potential impact of the pandemic on health policy understanding and the impact of an individual’s health status on the research results. We asked respondents to provide personal information related to their health status, including their health status during the pandemic and the number of uncomfortable days in the past year. We thus collected the following background information from respondents: gender (female = 1; male = 0); age (five levels, under 19 years = 1, to 5 for over 50 years); education level (five levels, from primary school = 1, to postgraduate and above = 5); average annual household income (five levels, under 80,000 CNY = 1, to 5 for over one million CNY); urban (urban resident = 1, Towns or rural and countryside = 0); Current health status better (Current health status very good or better = 1, Current health status bad or with chronic illness = 0); and feeling unwell for less than 15 days (feeling unwell for less than 15 days = 0, otherwise = 1).

### 3.3. Data Analysis Approach

As stated above, respondents’ level of understanding of the HCI is a five-scale ordered variable (no hear = 1, not understand = 2, midpoint = 3, understand = 4, very understand = 5) which does not satisfy the conditional constraints of a general linear regression. Therefore, we used ordered logistic regression to evaluate these relationships. In order to test the relationship between the independent variables and respondents’ understanding of the HCI, we constructed ordered logistic regressions with control variables.

Four independent regression models were evaluated to explore the relationship between smart healthcare and awareness of the HCI in order to test hypothesis 1. Model 1 regressed respondents’ familiarity with the concept of smart healthcare on the HCI. Model 2 adds all individual-level control variables (respondent demographic information and health status information) to Model 1. Models 3 and 4 were the robustness tests for hypothesis 1. In Model 3, the four questions related to smart healthcare were regressed on the HCI and used to compare consistency with the concept of smart healthcare. Finally, in Model 4, all individual-level control variables were added to Model 3.

## 4. Results

### 4.1. Overview of HCI Understanding Level

The results of the questionnaire showed that only 1.2% of people stated that they understood HCI very well, while 71.26% of respondents stated that they did not understand HCI. Further analysis showed that respondents who did not understand the HCI were mainly female, older, and less educated. A regional breakdown of the respondents’ places of residence also revealed significant differences in levels of understanding of the HCI by region, with nearly 73.1% of respondents who said they did not understand the HCI coming from the northern regions of China. Respondents from the eastern, southwestern, and northwestern regions, on the other hand, had a better understanding of the HCI.

### 4.2. Overview of Smart Healthcare Familiarity Level

As Table 3 shows, the Chinese public’s overall familiarity with the concept of smart healthcare is below the mid-point with a mean score of 2.66 (SD = 1.18) but higher than the Chinese public’s understanding of the HCI with a mean score of 2.20 (SD = 0.93). This indicates that respondents are significantly more familiar with the overall concept of smart healthcare than the HCI and that respondents rated it as predominantly more familiar.

We also surveyed the Chinese public’s familiarity with four questions related to smart healthcare, as shown in Table 3. These results revealed that the Chinese public was more familiar with disease prevention and risk monitoring than the other three smart healthcare applications, with a mean score of 3.23 (SD = 1.04). They also revealed that the Chinese public was least familiar with assisting diagnosis and treatment, with a mean score of 2.70 (SD = 0.91). This indicates that the respondents’ perception of smart healthcare is not just conceptual and that they possess a degree of understanding of the specific manifestations of smart healthcare. While the majority of respondents were more familiar with disease prevention and risk monitoring, a minority of respondents reflected a good understanding of assisting diagnosis and treatment.

### 4.3. The Relationship between the HCI and Smart Healthcare

As shown in Table 4, the empirical results are consistent with our research hypothesis, and our experiment confirms that the Chinese public’s familiarity with smart healthcare can significantly influence the degree to which people understand HCI. Model 1 shows a McFadden R-squared of 0.092 for overall conceptual familiarity with smart healthcare and a comparison of Models 1 and 2, with Model 2 showing a McFadden R-squared of 0.109 for individual-level demographic information and health status information. The existing research suggests that the larger the McFadden R-squared, the higher the explanatory power of the model [47]. Therefore, this result indicates that the overall concept of smart healthcare has a much higher explanatory power when it comes to how well members of the public understand the HCI than factors related to individual-level demographic information and health status information. The McFadden R-squared comparison between Model 2 (McFadden R-squared = 0.109) and Model 4 (McFadden R-squared = 0.141) further indicates that the four questions linked to smart healthcare have a relatively high explanatory capacity for the level of understanding of the HCI compared to the overall concept of smart healthcare.

In Model 2, all else being equal, a one-unit increase in respondents’ familiarity with smart healthcare was associated with a 1.30-fold (exp(0.259)) increase in understanding of the HCI. Similarly, in Model 4, except for the Chinese public’s familiarity with disease prevention and risk monitoring, which was not correlated with an understanding of the HCI, the other three items were significantly correlated with an understanding of the HCI. In particular, with all other things being equal, a one-unit increase in the respondents’ familiarity with assisting diagnosis and treatment was associated with a 1.44-fold (exp(0.364)) increase in understanding of the HCI. These results suggest that familiarity with smart healthcare positively impacts the level of understanding of the HCI, which is consistent with our hypothesis.

### 4.4. Impact of Demographic Factors on HCI

In the discussion above, we identified significant differences in gender, age, geography, and educational level that arose when respondents reported how well they understood the HCI. Our research further reveals some of the effects that these demographic factors (control variables) had on HCI understanding through regression.

According to Model 4, the statistics on demographic information variables showed that female respondents were 1.14 times (exp(0.134)) less likely to understand the HCI than male respondents. Older people were 1.31 times less understanding than younger people. In addition, income and education levels affected the respondents’ level of understanding. However, we did not find strong evidence to support a correlation between urban residents and high comprehension levels of the HCI.

### 4.5. Impact of Health Information on HCI

Our research found a significant correlation between the respondents’ health information and their level of understanding of the HCI. Specifically, the results depicted in Table 4 show that respondents with current health status better were 1.53 times (exp(0.423)) more likely to understand the HCI compared to respondents with poorer health status. Similarly, respondents who felt unwell for less than 15 days were more understanding of the HCI than respondents who felt unwell for more than 15 days. Thus, both the respondents’ current health status and long-term health status significantly influenced how well they understood the HCI.

### 4.6. Respondents’ Channels of Access to Health Information

Figure 1 shows the word cloud of these open-ended questions. The word frequencies for Chinese public opinion in the four open-ended questions can be viewed on the webpage (https://github.com/Winckwu/Reflecting-on-HCI-in-China, (accessed on 1 January 2023)). The results of the open-ended questionnaire related to methods of accessing health information are shown in Figure 1a, where the main knowledge channels are electronic media sources, such as the internet and television. However, a large proportion of respondents still obtain information through direct communication with their doctor. In contrast, social media platforms such as Douyin (TikTok) are not the most important channels for obtaining health information.

### 4.7. Respondents’ Topics of Concern

Figure 1b shows the diseases and other health problems about which the respondents were most worried. These mainly included chronic diseases, such as arthritis, high blood pressure, heart disease, diabetes, and asthma. The main medical treatment issues that respondents were concerned about were long waiting times, high costs, and the possibility of receiving incorrect or excessive treatment. In terms of the future of healthcare, respondents hoped that the Chinese government would focus on the development of the health industry, the promotion of health in age, and health management. Respondents had limited interest in the matters of medical check-ups, the pharmaceutical industry, and medical tourism.

## 5. Discussion

### 5.1. HCI Promotion Is Insufficient

Although studies show the role of public health policy in maintaining and promoting emotional well-being, our survey results show that more than 70% of respondents indicated that they were not aware of the HCI one year after its launch, a significant expected figure. China has a top-down approach to policy dissemination, which means that promotion strategies take time to move from development by higher levels of government to implementation by local governments. However, it is important to emphasize that the policy is not sufficiently publicized because the goals of HCI depend on public participation. For example, the policy states that by 2022, China should have a national health literacy level higher than 22%. Accomplishing this goal requires early public awareness of the policy and a significant period of time for people to develop and adopt the health literacy advocated by the HCI. However, one year after the policy’s implementation, 21.7% of respondents still had not heard of the policy, making it more difficult to achieve the 2022 health literacy goal and other key objectives of the plan. Several studies have concluded that China is falling short of achieving the goals set out in Health China 2030 [48]. In order to achieve the goals of the policy, public understanding, and support for the policy, especially in terms of health literacy, is critical.

#### 5.1.1. Community Involvement in Policy Promotion

There is a gap in HCI coverage that is hard to attribute to any one region. The results show that although respondents in Northwest China were more aware of HCI than other regions, over 60% of respondents in that region still reported a lack of knowledge about the policy. Thus, it is clear that there is a general lack of awareness of HCI policy. As we mentioned earlier, the Chinese government is currently actively promoting the policy on various platforms. The government is primarily responsible for the promotion of the policy, and according to the HCI website, approximately 20 government agencies are currently involved in the promotion of HCI [1]. This suggests that HCI promotion in China is currently dependent on government efforts. However, in a public emergency such as the COVID-19 pandemic, the government’s ability to accomplish this task would be severely compromised. The government could consider harnessing the potential of civil society organizations and social groups, drawing on the US experience, to encourage their participation and outreach.

#### 5.1.2. Smart Healthcare Improves Access to Information

Smart healthcare is a relatively familiar concept to the Chinese public, with people more familiar with disease monitoring and therapies. This may be related to the widespread use of smart medical technology during the pandemic, as previous studies found that public acceptance of online treatment increased significantly after the COVID-19 crisis. In addition, China requires all citizens to use mobile phone-based smart medical technology to help the Chinese Center for Disease Control (CCDC) locate and screen COVID-19 cases [49]. However, the widespread and mandatory use of this technology has the potential to violate the privacy of cell phone users [50]. On the other hand, it effectively helps CCDC to quickly target and block the areas where cases are located and slow down the spread of the epidemic [51]. These technologies indirectly make Chinese cell phone users more aware of smart healthcare. Our study shows that the participants most familiar with smart healthcare also tend to be more knowledgeable about HCI, which may provide insights for future HCI promotion. The promotion of health policies in China could also be used to popularize other policies related to smart healthcare.

#### 5.1.3. Recommendations in Relation to Policy Promotion

The internet has been the main platform used by the Chinese public to seek health advice. This may be due to the development and spread of electronic technology, which allows the public to access health advice directly online in a relatively convenient manner [52]. Meanwhile, travel restrictions imposed in China during the pandemic may have affected public access to information [53]. However, traditional ways of accessing health information, such as medical consultations, books, and television, remain important channels for respondents to obtain health information. Therefore, the government should use multiple channels to disseminate health information and health policies. Meanwhile, the Chinese public remains concerned about a number of chronic diseases, many of which have been included in the policy objectives of the HCI, such as diabetes, heart disease, and respiratory disease. This suggests that the HCI was developed with a good understanding of the health status of the Chinese public. Diseases of particular public concern could also be highlighted in the policy communication process to achieve more effective policy dissemination.

### 5.2. Limitations and Further Directions

Although we have tried to overcome limitations, every research study has challenges. This study collected data one year after the policy was released, which does not provide a complete picture of the public’s reaction to the policy, as many of its promotional measures may not have yet been fully implemented. Furthermore, this study collected data one year after the policy was released, which does not fully reflect the public’s response to the policy because many of its promotional measures may not have been fully implemented, and views toward it may have changed in light of recent events, implying that more research on the extent of public awareness of HCI is required. In addition, because we relied primarily on an online questionnaire, our study may have excluded some users who do not use social media. Finally, future research could also look at the promotion of some policies related to healthcare topics that our study found to be of particular interest to the public, such as healthcare reform policies, healthcare industry policies, and smart healthcare support policies.

## 6. Conclusions

Numerous nations have long-term health policies, and the promotion of these policies plays a crucial role in accomplishing major objectives. Nevertheless, each nation has a unique strategy for promoting policies. Our study focuses primarily on the implementation of China’s long-term health strategy, the HCI, and the implementation policy of Healthy China 2030. As a long-term national policy, there is no doubt that the HCI will have tremendous effects on all Chinese residents. China is still in the early stages of the Healthy China strategy’s transformation, and healthcare reform is in full gear, yet revolutionary reforms always have short. It is anticipated that there will be a processing deficit of medical resources as a result of efforts to discard the old system and traditional procedures and develop a new medical system. This trend was exacerbated by the big COVID-19 outbreak in 2020–2022, with fluctuating infection rates across the nation and recurrent strains on medical resources everywhere. However, the short-term effects of adopting long-term health measures reduce public anxiety.

By developing a long-term health policy, the short-term effects generated alleviate the uncertainty caused by the pandemic and, thus, the anxiety caused by public fear [31]. However, public awareness of long-term health policies is crucial in this case [6], and this effect can be effectively enhanced by raising health awareness. This paper finds that the dissemination and application of smart health technologies can educate the public about long-term health policies along with information technology, thus demonstrating that when the details of a policy become more concrete and clearer, the public can become more aware of it, thus alleviating uncertainty about unknown environmental threats, and further maintaining and promoting positive emotions. Thus, the availability of advanced health-related technologies can help improve the diffusion of health policy and provide new insights to policymakers and participants. It can also provide lessons for other countries in the early stages of policy diffusion, particularly health policy advocacy and promotion during COVID-19. We also found that using the Internet as an information platform for policy promotion helped to reach a wider public and thus could play a central role in future health policy diffusion efforts in China, where the Internet is well integrated into society.

However, with the mutation of the virus strain and its increased potential to escape, the urge to adopt high-intensity precautions against it led to the disruption of productive life and the gradual rise of opposition and protests. In late 2022, China classified COVID-19 as a category B infectious disease and gradually took liberalization measures. As infection rates soared, medical resource bottlenecks occurred at an even faster pace. However, during the past two years, the Chinese government has continued to popularize national vaccination against COVID-19 aggressively as a preventive measure aimed at avoiding infection and mitigating symptoms of infection. According to data published by the National Health Commission, on the eve of liberalization, the complete vaccination rate for the entire population in China exceeded 90%, and the state has attempted to prevent medical crowding out, which has directly reduced the rate of severe infections after liberalization. So, the gradual liberalization, which is still based on long-term health policy coordinated with economic development, has the long-term effect of alleviating the social, economic, and political inequalities caused by the long-term effect of this approach is to alleviate health inequities induced by social, economic, and political inequality. However, the emergence of opposition voices and the findings of this study remind us that in the future, we must pay greater attention to policy publicity, interpretation, and public popularization, which can also serve as an example for other nations in the initial stages of policy dissemination, particularly health policy advocacy and promotion during epidemics.

## Figures and Tables

**Figure 1 ijerph-20-02889-f001:**
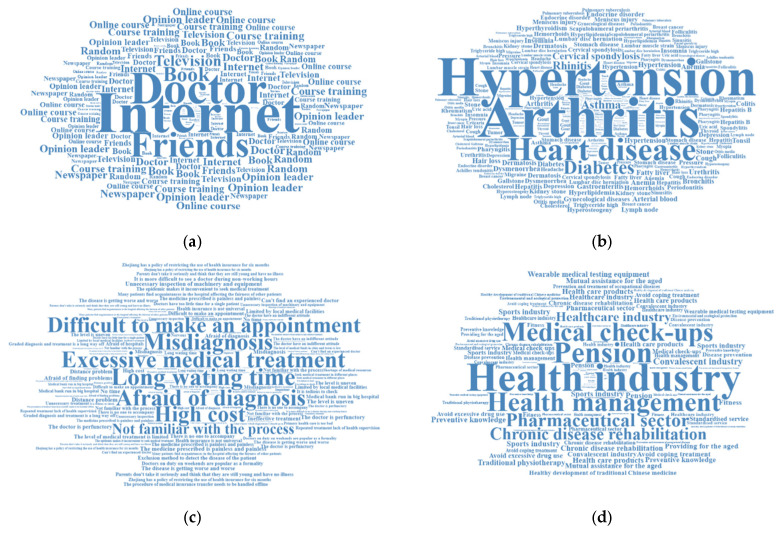
Word cloud of Chinese public opinion in four open-ended questions. (**a**) Channels to access health information. (**b**) Diseases and health problems of greatest concern. (**c**) Medical treatment issues of greatest concern. (**d**) Health areas that government should prioritize in the future.

**Table 1 ijerph-20-02889-t001:** Summary of China’s long-term health policy from 2009 to now.

Policy/Program	Policy Publisher(s)	Objectives
Healthy China 2020 program (2009)	Ministry of Health (Now NHC)	Further improve key national health indicators so that, by 2020, life expectancy per capita will reach 77 years, under-five mortality will fall to 13 per thousand, maternal mortality will be reduced to 20 per 100,000, and disparities in health status between regions will be reduced. Improve the accessibility and equity of health services. Reduce the economic risk of illness among the population. Control risk factors to halt, reverse and reduce the spread of chronic diseases and health hazards. Strengthen the prevention and control of infectious and endemic diseases and reduce the risks associated with them. Strengthen monitoring and supervision to ensure food and drug safety. Embrace scientific and technological progress, adapting to the transformation of medicine and realizing a forward-focused and integrated strategy. Adapt and update traditional Chinese medicine and utilize it as much as possible in the safeguarding of national health. Develop the health industry to meet the demand for multi-level and diversified health services. Increase investment in health so that, by 2020, the proportion of total health expenditure to GDP will grow from 6.5% to 7%, and the strategic goal of Health China 2020 will be achieved.
Healthy China 2030 Planning Outline (2016)	Central Committee of the Chinese Communist Party and State Council of the People’s Republic of China (PRC)	A high-quality, efficient, and integrated medical and health system will be fully established, along with a comprehensive public system for national fitness; the health protection system will be further improved; the overall strength of health science and technology innovation will rank among the world’s foremost; and the quality and level of health services will be significantly improved. A health industry with an optimized structure will be established, and a number of large enterprises with strong innovation capability and international competitiveness will be formed, becoming a pillar of the national economy. The system of policies and laws conducive to health will be further improved, and the system of governance and governance capacity in the field of health will be modernized.
HCI (2019–2030) Action plan for Healthy China 2030 policy	State Council of the PRC	Implement health literacy initiatives. Promote a healthy diet. Implement a national fitness campaign. Implement tobacco control initiatives. Promote mental health and well-being. Promote the development of health services. Promote maternal and child health. Oversee the promotion of health activities in primary and secondary schools. Promote health in old age. Promote occupational health. Take action to support the prevention and treatment of cardiovascular and cerebrovascular diseases. Take action to support the prevention and control of cancer. Take action to support the prevention and treatment of chronic respiratory diseases. Take action to support the prevention and control of diabetes. Take action to support the prevention and control of infectious and endemic diseases.

**Table 2 ijerph-20-02889-t002:** Participants’ demographic information.

Parameter		China Statistical Yearbook 2020	Research Sample	Num. of Cases
Gender	Female	48.91%	47.11%	1172
Location	Northern	12.51%	13.87%	345
	Northeastern	7.69%	6.15%	153
	Eastern	29.48%	34.81%	866
	Central & Southern	28.43%	25.40%	632
	Southwestern	14.50%	13.22%	329
	Northwestern	7.38%	6.55%	163
Age (18 years or older)	≤19	3.49%	2.85%	71
	20–29	16.21%	18.45%	459
	30–39	19.43%	21.26%	529
	40–49	19.54%	22.63%	563
	≥50	41.33%	34.81%	866
	Average	42.4	41.31	/
Income	Average disposable household income	95,672.8 CNY	96,298 CNY	/
Ethnicity	Han	91.11%	93.77%	2333

Source: China Statistical Yearbook 2020.

**Table 3 ijerph-20-02889-t003:** Descriptive Statistics.

Variables	Mean	Std Dev	Min	Max
**Dependent variables**				
HCI	2.20	0.93	1	5
**Explanatory variables—Group 1**				
Smart Healthcare	2.66	1.18	1	5
**Explanatory variables—Group 2**				
Assisting diagnosis and treatment	2.70	0.91	1	5
Health management	3.08	1.10	1	5
Disease prevention and risk monitoring	3.23	1.04	1	5
Assisting drug research	3.11	0.94	1	5
**Control variables**				
**Demographic factors**				
Female	0.44	0.50	0	1
Age	2.64	0.90	1	5
Education level	3.52	1.04	1	5
Urban	0.62	0.42	0	1
Average annual household income	2.56	1.13	1	5
**Health status information**				
Current health status better	0.81	0.39	0	1
Feeling unwell for less than 15 days	0.21	0.41	0	1

**Table 4 ijerph-20-02889-t004:** Ordered Logit Model of HCI understanding level.

	Model 1	Model 2	Model 3	Model 4
**Variables**				
**Group 1**				
Smart Healthcare	0.251 *** (0.032)	0.259 *** (0.033)		
**Group 2**				
Assisting diagnosis and treatment			0.270 *** (0.042)	0.364 *** (0.043)
Health management			0.080 ** (0.041)	0.103 *** (0.042)
Disease prevention and risk monitoring			−0.004 (0.042)	0.028 (0.042)
Assisting drug research			0.210 *** (0.045)	0.230 *** (0.046)
**Demographic factors**				
Female		−0.176 ** (0.080)		−0.134 ** (0.080)
Age		0.323 *** (0.046)		0.458 *** (0.048)
Education level		0.144 *** (0.038)		0.107 *** (0.038)
Urban		−0.169 (0.102)		−0.236 (0.103)
Average annual household income		−0.066 * (0.038)		−0.133 *** (0.037)
**Health status information**				
Current health status better		0.442 *** (0.104)		0.423 *** (0.105)
Feeling unwell for less than 15 days		0.288 *** (0.100)		0.408 *** (0.101)
N	2488	2488	2488	2488
McFadden R-squared	0.092	0.109	0.124	0.141

Notes: Standard errors in parentheses; *** *p* < 0.01, ** *p* < 0.05, * *p* < 0.1; constant cuts omitted.

## Data Availability

The data presented in this study are available in https://github.com/Winckwu/Reflecting-on-HCI-in-China (accessed on 1 January 2023).

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
