# Peer review of "Insights from COVID-19: Reflecting on the Promotion of Long-Term Health Policies in China"

_ijerph, 2023, doi:10.3390/ijerph20042889_

Round 1

Reviewer 1 Report

Comments to authors

This is a nice paper. I would suggest trying to edit it down a bit. Also be clear about its focus: COVID is used as an example, but the paper is really about knowledge of China Long-term Health Policy.

line 82-please clarify what you mean by "virtuous cycle" is this a quote? To what are you referring?

line 86 Health People 2030 US was launched Aug 18,2020, you should refer to and cite this instead of 2020.

line 185 insert "of" before the word "time"

line 204  what are you trying to communicate with phrase "people do not create a vacuum in the system"--this makes the sentence unclear.

line 219 refers to "effective containment' of COVID, in light of recent events insert a date before this.

line 226-Please give a full reference for the assertion "all 60 new cases" came from abroad. You site WHO but it would be relying on China reports. Also what was the travel policy at the time. Are you instead meaning to say "of the 60 newly reported cases, all were associated with X number of cases from abroad" Is it possible 60 people entered testing negative and converted in China unrelated to any transmission to them in China?

line 238 Table 2 -Add Ns to all percentages list (i.e.number of cases)

HCI is listed as the singular dependant variable. Discuss in the end how views toward it may have changed in light of recent events--might they undermine faith in the "long term health policy" and suggest further research

line 253 further define "self-designed "by authors" so there is no confusion

265 you describe the weChat survey as "privacy-free and anonymized" This is confusing. Are you saying you knew the identity of respondents and then de-identified? This could introduce bias particularly with regard to health condition disclosure. In any event describe what you did better. 

Do you have penetration of wechat use in China by age etc--if so cite it

Line 282 what is a "red packet", assume the survey with some incentive. Explain it more clearly

line 289 define "square regression analysis" do you mean "least squares"

line 297-307 Consider making clear how your AI analysis ruled out the appearance of words that had modifiers in front of them that changes the meaning. As an example GOOD with "not" before it.

I think you might consider handling the word cloud data in a narrative form and drop the tables but live link them them for interested readers who want to view

line 324 the data were collected by age category correct? or did people state age and you put into categories? say which

line 328 Define "day to day basis" the first time you use it

line 351-352 sentence poorly worded

line 411-412 reword "less likely to understand the HCI"

line 469-476 there is redundancy verbatim with the introduction

line 478-479 what do mean by "promoting emotion'? do you mean to say :emotional well-being"?

line 509-510 you say population most familiar with "complementary diagnostics.." I thought  earlier you said it was "disease monitoring"

The conclusion should make reference to the recent China surge in cases with the change in policy demanded by protestors and the need to conduct research on the impact of these events on HCI understanding and support.

Author Response

Dear reviewer,

Thank you so much for giving us another chance to revise and resubmit the manuscript. We have paid close attention to the issues raised by both reviewers, especially in introducing the theoretical perspective of the study.

The attachment present our detailed responses to the comments by you. We hope that the revised version of the manuscript could meet with your final approval.

Thank you.

Best regards

Qi

Reviewer 2 Report

Abstract: it is unstructured, it should stick to journal requirements. There should be clearly stated that you did the survey and what are results of it are.

Introduction

Please add information that more  about the HCI policy will be in the literature review part.

Literature review

I do not understand why Hypothesis 1 appeared at such a moment, not in the methods section. Please delete this hypothesis from here, and do you need the word can inside? If there is only one hypothesis it does not need the number.

Methods

Is your database publicly available?

Results

You did not introduce the variables from table 3 before it, and I did not understand them and their numbers while reading for the first time. What scale was used, and what mean means.

“The actual value was RMB 1 per person.” How much is it in euros?

Another time we have table 3. Right now it is more understandable.

Point 4.5  I do not think there is a sense of showing all these tables. The word clouds are enough. The questions should be repeated in their original version, to understand this part well.

 Limitations

Your study was not representative – this is the main limitation.

 Lines 543 to 547 are not limitations. The results and its distribution are not limitation.

Conclusion

I would reconsider the conclusion if they really reflect what was discovered in the article.

Where is appendix 1?

Author Response

Dear reviewer,

Thank you so much for giving us another chance to revise and resubmit the manuscript. We have paid close attention to the issues raised by both reviewers, especially in introducing the theoretical perspective of the study.

The attachment presents our detailed responses to the comments by you. We hope that the revised version of the manuscript could meet with your final approval.

Thank you.

Best regards

Qi

Round 2

Reviewer 2 Report

No further comments.